# Fish mislabelling in France: substitution rates and retail types

Julien Bénard-Capelle[1], Victoire Guillonneau[2], Claire Nouvian[2], Nicolas Fournier[3], Karine Le Loët[4] and Agnès Dettai[5]

[1] Institut National de la Santé et de la Recherche Medicale, Unité 1001 Robustness and Evolvability of Life, Université Paris Descartes Sorbonne Paris Cité, Paris, France
[2] BLOOM, Paris, France
[3] Oceana Europe, Brussels, Belgium
[4] Terra Eco, Nantes, France
[5] Département Systématique et Évolution, Muséum national d'Histoire naturelle, Sorbonne Universités, UMR 7205 MNHN-CNRS-UPMC-EPHE, Paris, France

## ABSTRACT

Market policies have profound implications for consumers as well as for the management of resources. One of the major concerns in fish trading is species mislabelling: the commercial name used does not correspond to the product, most often because the product is in fact a cheaper or a more easily available species. Substitution rates depend heavily on species, some often being sold mislabelled while others rarely or never mislabelled. Rates also vary largely depending on countries. In this study, we analyse the first market-wide dataset collected for France, the largest sea food market in Europe, for fish species substitution. We sequenced and analysed 371 samples bearing 55 commercial species names, collected in fishmonger shops, supermarkets and restaurants; the largest dataset assembled to date in an European country. Sampling included fish fillets, both fresh and frozen, and prepared meals. We found a total of 14 cases of mislabelling in five species: bluefin tuna, cod, yellowfin tuna, sole and seabream, setting the overall substitution rate at 3.7% CI [2.2–6.4], one of the lowest observed for comparable surveys with large sampling. We detected no case of species mislabelling among the frozen fillets or in industrially prepared meals, and all the substitutions were observed in products sold in fishmongers shops or restaurants. The rate of mislabelling does not differ between species, except for bluefin tuna. Despite a very small sample size ($n = 6$), the rate observed for this species (83.3% CI [36–99]) stands in sharp contrast with the low substitution rate observed for the other substituted species. In agreement with studies from other countries, this work shows that fish mislabelling can vary greatly within a country depending on the species. It further suggests that more efforts should be directed to the control of high value species like bluefin tuna.

Subjects Aquaculture, Fisheries and Fish Science, Food Science and Technology
Keywords Fish, Mislabelling, Species substitution, France, DNA barcoding, Retail, Bluefin tuna, Citizen science

## INTRODUCTION

Fish species represent an important and globally growing food resource. Most of the fish supply is harvested in the wild, as aquaculture represents just over 42% of the

Corresponding author
Agnès Dettai, adettai@mnhn.fr

fish consumed in the world (*FAO, 2014*) and only slightly more than 30% for France (*Meunier, Daurès & Girard, 2013*). An efficient management of these natural resources is particularly important, as currently 29.9% of fish stocks are overexploited, 57.4% are fully exploited, and 7.6% of global production comes from stocks that are collapsed or recovering (*FAO, 2012*). However, the management of wild marine resources is adversely affected by unreliable traceability and labelling, impeded by the wide trade of aquatic food (*Cochrane et al., 2009*). A wide array of species coming from geographically distant areas is now available in many markets, making effective control along the supply chain complex.

In Europe, a number of policies regulate seafood labelling and traceability ((*EUR-Lex, 2014*) 104/2000, 2065/2001, 178/2002, 1224/2009). Efforts have also been directed towards consumers to drive demand towards species with less conservation issues. In France as in other countries, fish buying guides are available to help customers choose among species depending on conservation issues, origin of the product, fishing methods or season (*Hanner et al., 2011*; *Jacquet & Pauly, 2008*).

Mislabelled fish may lead customers to unknowingly purchase products not corresponding to their ethical or taste criteria (*Rasmussen & Morrissey, 2008*; *Jacquet & Pauly, 2008*). When mislabelling is discovered and made public, it may reduce trust between consumers and suppliers. Moreover, some species or provenances can even represent health hazards (*Jacquet & Pauly, 2008*). The species designation is therefore crucial in allowing an informed choice by consumers, and needs to be reliable and correct. In Europe, fish is second on the list of products that are the most at risk from food fraud, and Europol has observed a rise in the number of general food fraud cases (*Committee on the Environment, Public Health and Food Safety, European Parliament, 2013*).

The advent of molecular identification methods has given rise to a scientific attempt to quantify the amount of seafood mislabelling. While protocols and methods are not yet standardized, these studies have shown that the situation varies greatly between products and countries (*Barbuto et al., 2010*; *Cutarelli et al., 2014*; *Di Pinto et al., 2013*; *Filonzi et al., 2010*; *FSAI, 2011*; *Garcia-Vazquez et al., 2011*; *Griffiths et al., 2013*; *Huxley-Jones et al., 2012*; *Lowenstein, Amato & Kolokotronis, 2009*; *Machado-Schiaffino, Martinez & Garcia-Vazquez, 2008*; *Miller & Mariani, 2010*; *Miller, Jessel & Mariani, 2012*; *Vinas & Tudela, 2009*). For instance, high rates of substitution have been observed in Italy for some species (77.8% for *Mustelus* sp., in *Barbuto et al. (2010)*, or 53.36% for cod in *Di Pinto et al., 2013*) while less than 1.5% substitutions were uncovered in industrially prepared food purchased in British supermarkets (*Huxley-Jones et al., 2012*). This first suggests that any attempt to quantify the rate of fish mislabelling in a new country or region must be done carefully, using a wide sampling as representative as possible of the market. This diversity of results also calls for a comparative analysis of different markets: understanding the ecological, cultural and economic grounds influencing mislabelling might help reduce it.

France is the largest seafood/fish market among European countries, with more than 2 millions tons consumed in the country, and is ranked 7th for per capita consumption (*FAOSTAT* data for year 2011), yet no results have been published on fish mislabelling.

Government agencies regularly control the quality of the seafood supply, but their conclusions are not available to the public (*DGCCRF, 2013*).

Food control, and the detection of species substitution in particular, has benefited greatly from the development of DNA-based methods to identify species food content when morphology cannot be used. These techniques provide cheap and fast identification with little need for initial knowledge of the samples. They have proved very useful for the identification of fish species (for a review, see *Griffiths et al., 2014*; *Rasmussen & Morrissey, 2008*). Among the markers used, mitochondrial DNA sequences have emerged as near-universal markers for precise determination of species. The most frequently used sequences are partial cytochrome b, partial 16S or 12S ribosomal DNA, and partial cytochrome oxidase I (COI). The DNA sequence is then compared to reference sequences to identify the taxonomic group. The development of online databases containing thousands of DNA sequences has further enhanced the reliability and ease of use of these methods. It is particularly important that the reference sequence for the searched species is present in the reference dataset, so more complete datasets are more valuable (*Ekrem, Willassen & Stur, 2007*). The Barcode of Life Database (BOLD, *Ratnasingham & Hebert, 2007*) currently contains almost 150,000 COI barcode sequences for almost 14,000 actinopterygian species. Cytochrome b, the second largest, only has around 82,000 sequences listed in the GenBank nucleotide database. Additional features of the Barcode of Life project, such as linking sequences to vouchered specimens and specimen data, increase reliability compared to the notoriously high error rates in GenBank (*Harris, 2003*; *Rasmussen & Morrissey, 2008*). With the success of the Barcode of Life Project for fish (*Ward, Hanner & Hebert, 2009*), its reference marker, COI, has been increasingly used for identifications and represents the majority of substitution studies in the last years. Other datasets for the regulatory identification of species substitutions have also been established for COI; for instance, in the Regulatory Fish Encyclopedia of the Food and Drug Administration (*Yancy et al., 2008*).

The present study, the first of its kind for France, therefore aims to evaluate the extent of the mismatch between the market names and the actual species for some of the most common commercial marine fish species in France, including Bar, Lieu noir, Cabillaud, Merlu, Lotte, Merlan, Sole, Pangas, Raie Thon and other less represented species. The difference between results by country and species hints at different and specific effects depending on the market. A study which widely sampled the diversity of fish products available to the customer in France was therefore required for a first assessment of substitution rates. We focused on the less recognizable products: fillets (both sold by fishmongers and industrially packaged or deep-frozen), and dishes (either ready-made or served in restaurants). These products are particularly susceptible to be substituted as the customers and control agencies cannot easily recognize the species from the appearance of the product. Two collecting efforts have been started in parallel by the NGO Oceana, associated with the magazine Terra Eco, and the NGO Bloom, in collaboration with researchers from the French National Institute of Health and Medical Research (INSERM) and National Museum of Natural History (MNHN). These two initiatives

took advantage of the development of citizen science to increase the sampling's coverage, both geographically and by place of purchase. They are both analysed and presented in this article, resulting in the largest European dataset for such a study.

## MATERIAL AND METHODS

### Sampling

Samples and corresponding data were collected across continental France between April and December 2013. Two independent sets of samples were collected, hereafter referred to as FishLabel (FL) and TerraEco (TE).

#### FishLabel set

The FL set was collected in pre-numbered tubes and stored in 95% ethanol until extraction. Each sample was divided at sampling in two tubes with the same sampling number. Data such as commercial name, Latin name when indicated, date, collector, location, brand name, shop or restaurant names were collected, as well as photographs of packages and samples when possible. The data were collected either on paper forms sent along with the samples or uploaded online using the smartphone application Epicollect (*Aanensen et al., 2009*). Only fillets or fish dishes were sampled as they are not readily identifiable and thus potentially easier to substitute. They were collected from fishmonger shops, restaurants, and supermarkets (either at the fishmonger department or industrially prepared, i.e., canned or fresh ready-made meals and deep-frozen fillets). To avoid dispersion over a large number of species, we focused the sampling on ten commercial names chosen among the most consumed fish species in France (according to www.franceagrimer.fr, checked April 2013): Bar or Loup, Lieu noir, Cabillaud, Merlu or Colin, Baudroie or Lotte, Merlan, Sole, Pangas, Raie, Thon (See Table 1 for correspondence with species names and English names). Although salmon is the most consumed fish in France, we initially excluded all salmon species from the list of targeted species because the market is dominated by cheap *Salmo salar* from aquaculture, which is expected to be less substituted because of its price. The detailed instructions to the samplers are provided online (in French and translated in English, FigShare http://dx.doi.org/10.6084/m9.figshare.978485). The collectors were contacted through the personal connections of the authors. They were provided with sampling kits containing ten collecting tubes and written detailed instructions, as well as a return envelope.

#### Terra Eco set

Detailed instructions to the samplers are provided online (in French and translated in English, FigShare http://dx.doi.org/10.6084/m9.figshare.978485), following the protocol used by Oceana in the USA. The collectors were asked to sample only the products labelled as Cabillaud (cod), Lotte (anglerfish) or Thon rouge (bluefin tuna). Only one tube was prepared for each sample, and they were conserved in silica gel until extraction.

### DNA extraction and sequencing

Prior to analysis, the samples that lacked crucial data (defined as collection site, retail name, dish name including species name and collector, and clear indication of the

**Table 1 Summary of the sampled species.** The samples are presented per protocol (FL or TE) and commercial name. When several species are sold under one name, the species present in the dataset according to molecular ID are highlighted in bold.

| Commercial name | English name | Accepted species[a] | Number of samples (substitution cases) | | | | | Total |
|---|---|---|---|---|---|---|---|---|
| | | | Fishmonger | Supermarket fresh filet | Deep-frozen filet | Restaurant | Ready-made dish | |
| **FL samples** | | | | | | | | |
| Thon* | Tuna | ***Thunnus albacares*, *others*** | 5 | 1 | | 8 | | 14 |
| Thon rouge | Atlantic bluefin tuna | ***Thunnus thynnus*** | 1(1) | | | 1(1) | | 2 |
| Thon germon | Albacore tuna | ***Thunnus alalunga*** | 1 | | | 1 | | 1 |
| Thon albacore | Yellowfin tuna | ***Thunnus albacares*** | 2 | 4(1) | 1 | 4 | 1 | 12 |
| Thon listao | Skipjack tuna | ***Katsuwonus pelamis*** | | | | 1 | 1 | 1 |
| Maquereau | Atlantic mackerel | ***Scomber scombrus*** | | | | 1 | 1 | 2 |
| Cabillaud, morue | Cod (both Atlantic and Pacific cod) | ***Gadus morhua*, *G. macrocephalus*,** *G. ogac, Arctogadus glacialis, Boreogadus saida, Eleginus navaga, E. gracilis* | 12 | 27(2) | 8 | 19(1) | 23 | 89 |
| Colin d'Alaska, Lieu d'Alaska | Alaska pollock | ***Gadus chalcogrammus*** (*Theragra chalcogramma*) | | 2 | 4 | | 27 | 33 |
| Lieu jaune | Pollack | ***Pollachius pollachius*** | 1 | | | 1 | | 2 |
| Lieu noir, lieu, colin (lieu) | Saithe | ***Pollachius virens*** | 1 | 12 | 1 | 3 | 3 | 20 |
| Aiglefin, églefin | Haddock | ***Melanogrammus aeglefinus*** | | 1 | | | 2 | 3 |
| Merlan | Whiting | ***Merlangius merlangus*** | 2 | 11 | | | 2 | 15 |
| Julienne | Ling | ***Molva molva*** | | 3 | | | | 3 |
| Lingue | Blue ling | ***Molva dypterygia*** | | 1 | | | | 1 |
| Merlu blanc, merlu blanc du Cap, merlu (restaurant) | Hake | *Merluccius capensis, M. bilinearis, M. cadenati, M. gayi gayi, M. albidus,* ***M. hubbsi*, *M. paradoxus*,** *M. polylepis, M. productus, M. senegalensis* | | | 2 | 1 | 5 | 8 |
| Merlu, colin | European hake | ***Merluccius merluccius*** | | 1 | | 2 | | 3 |
| Grenadier | Roundnose grenadier | ***Coryphaenoides rupestris*** | | 2 | | | | 2 |
| Lotte | Anglerfish | *Lophius americanus,* ***L. piscatorius*,** ***L. budegassa*** | 1 | 1 | | 4 | | 6 |
| Lotte du Cap | Devil anglerfish | ***Lophius vomerinus*** | | | 1 | | | 1 |

| Commercial name | English name | Accepted species[a] | Number of samples (substitution cases) | | | | | |
|---|---|---|---|---|---|---|---|---|
| | | | Fishmonger | Supermarket fresh filet | Deep-frozen filet | Restaurant | Ready-made dish | *Total* |
| Bar, Loup | | *Dicentrarchus labrax* | 2 | 2 | | 11 | | 15 |
| Sole | Common sole | *Solea solea* | | 2(1) | | 5 | | 7 |
| Flétan | Atlantic halibut | *Hippoglossus hippoglossus*, H. stenolepis | | 2 | | | | 2 |
| Turbot | Turbot | *Scophthalmus maximus* | | | | 2 | | 2 |
| Limande du Nord | Yellowfin sole | *Limanda aspera* | | | | | 1 | 1 |
| Espadon | Swordfish | *Xiphias gladius* | | 1 | | 4 | | 5 |
| Daurade royale | Gilthead seabream | *Sparus aurata* | | | | 4 | | 4 |
| Sébaste[a] | Rockfish | *Sebastes sp.* | 1 | | | | | 1 |
| Pagre (restaurant) | Red porgy | *Pagrus pagrus, Pagrus sp.* | | | | 1(1) | | 1 |
| Loup de mer | Wolffish | *Anarhichas denticulatus, A. lupus, A. minor* | | 1 | | 1 | | 2 |
| Sabre noir | Black scabbardfish | *Aphanopus carbo* | | 3 | | | | 3 |
| Saumon, saumon atlantique | Salmon | *Salmo salar* | | | | 4 | 1 | 5 |
| Panga | Panga | *Pangasius hypophthalmus* | | 4 | | | | 4 |
| Raie | Ray | *Leucoraja naevus, Bathyraja brachyurops* + 25 species | 1 | | 1 | 1 | | 3 |
| Requin peau bleue | Blue shark | *Prionace glauca* | | 2 | | | | 2 |
| Rouget | | *Mullus surmuletus* | | | | 1 | | 1 |
| | | | | | | | Total | 276 |
| **TE samples** | | | | | | | | |
| Thon | Tuna | | 2 | | | 15 | | 17 |
| Thon albacore | Yellowfin tuna | *Thunnus albacares* | | 1 | | | | 1 |
| Thon rouge | Atlantic bluefin tuna | *Thunnus thynnus* | 4(3) | | | | | 4 |
| Lotte | Anglerfish | *Lophius americanus, L. piscatorius, L. budegassa* | 18 | 2 | | | | 20 |
| Cabillaud, morue | Cod (both Atlantic and Pacific cod) | *Gadus morhua, G. macro-cephalus, G. ogac, Arctogadus glacialis, Boreogadus saida, Eleginus navaga, E. gracilis* | 20(3) | 12 | 15 | 6 | | 53 |
| | | | | | | | Total | 95 |

**Notes.**

Table assembled using http://www.economie.gouv.fr/dgccrf/Poissons. Numbers in brackets are the number of substitutions after sample sorting.

[a] No qualifier is needed for restaurants for the commercial denomination to be correct.

sample number) were excluded. The FL samples were extracted using an epMotion 5070 (Eppendorf) and Tissue extraction kits (Macherey Nagel) following the instructions from the manufacturer. For practical reasons, samples that arrived later were extracted following the protocol in *Winnepenninckx, Backeljau & De Wachter (1993)*.

The partial COI was amplified using the primers FishF1, FishF2 and FishR1 from *Ward, Hanner & Hebert (2009)* and TelF1 and TelR1 from *Dettai et al. (2011)*. Samples with denatured DNA could not be successfully amplified, and the published primers for short fragments of amplified regions gave no variability between most *Thunnus* species. New primers were therefore designed that flanked variable areas and diagnostic sites of the *Thunnus* sequences (*Lowenstein, Amato & Kolokotronis, 2009*): COIF268-5′ GAAACTGACTYATTCCTYTAATGAT3′, COIF270-5′ AACTGACTTATTCCYYTAAT- GATYGG 3′, COIR450-5′ GAAGTTAATTGCCCCAAGAATTGA 3′, and COIR445-5′ AAGTTAATTGCTCCAAGAATTGAWGA 3′. Combinations of FishF1 or TelF1 and FishR1 or TelR1 produced a fragment of 652 bp, the primer couples COIF268-FishR1 or COIF268-TelR1 produced intermediate-sized fragment of 442 bp, and COIF268-COIR450 produced 208 bp fragments. All samples were first tested with the primers for the longest fragment. If this PCR was unsuccessful, we tested the intermediate size, and finally the shortest fragment. PCR followed *Dettai et al. (2011)* on Biorad thermocyclers. Purification and sequencing of the PCR products were performed commercially by GATC (http://www.gatc-biotech.com/) using the same primers. Most sequences were obtained in only one direction, but as a precaution, 70 samples chosen at random were sequenced in both directions. Samples where molecular identification differed from the commercial label were extracted from the second sample tube using the protocol in *Winnepenninckx, Backeljau & De Wachter (1993)*, amplified and sequenced a second time, when possible with a different pair of primers. Sequences were checked manually against their chromatogram using a Codoncode Aligner (CodonCode Corporation) and then exported and aligned in Bioedit (*Hall, 1999*).

The TE samples were extracted, amplified and sequenced with the same primers by Spygen, a commercial company specialised in molecular identification (http://www. spygen.fr/). The sequences provided were analysed by the same person, and using the same approaches as the FL dataset.

Sample descriptions and sequences are available in the Barcode of Life Database in the FSCF project (FCSF001-14 to FCSF291-14 for the FL dataset, FCSF292-14 to FCSF404-14 for the TE dataset), and in GenBank. Collector names, brands and precise collection data were anonymized. Photographs are included for samples when they do not threaten the anonymity of the data.

## Molecular identification

Three datasets were assembled according to the length of the sequences obtained (long, intermediate or short sized fragments). Within each dataset, pairwise distance trees were built with the taxon ID tree function included in BOLD to cluster identical sequences. These sequences were grouped in the alignment files, and sequence identities were also

checked on the alignments. Each distinct sequence was then used to BLAST-search the Barcode of Life database. The long dataset was compared to the Species Level Barcode Records, while the medium and short datasets were matched to the Full Length Record Barcode Database to avoid issues due to insufficient overlap of the sequences with the reference dataset. Identification was determined by sequence similarity to the reference dataset (*Wong & Hanner, 2008*), and checked through their position (*Costa et al., 2012*) in the "Tree based identification" generated distance trees in BOLD. For species with low interspecific divergences (*Gadus* and *Thunnus* species), aligned sequences were compared to each other, to sequences from the BOLD, and to sequences from the FDA reference dataset for Seafood identification (http://www.fda.gov/Food/FoodScienceResearch/DNASeafoodIdentification/ucm238880.htm). Additionally, we checked species-specific characteristic attributes and characteristic combinations on the alignments following *Lowenstein, Amato & Kolokotronis (2009)*.

## Mislabelling determination

For each sample, the list of admissible species that can be sold under the commercial name indicated on the menu, the price tag, or the box was determined by consulting a governmental website (http://www.economie.gouv.fr/dgccrf/Poissons, last checked on 25/02/2014). The sample was declared mislabelled if the species name determined through molecular identification was not in this list.

We did not retain the commercial names obtained orally from the waiting staff in the calculations of the substitution rates. However, we have kept this information in the data files.

## Grouping of commercial names

The total number of commercial denominations retrieved from the completed forms was high (55 different commercial names), preventing statistical analysis of a large part of the dataset. The samples were thus grouped into broader commercial categories. For instance "cabillaud" (cod) was grouped with "cabillaud du pacifique" (pacific cod) and "morue" (a French nomenclature for dry and salted cod, whether Pacific or Atlantic) under "cabillaud." This case and others like it reduced the number of categories to 30. We further decreased the number of categories by keeping only those for which at least 10 samples were available. All the other samples were grouped under the "other" category. However, after a preliminary analysis, it appeared that the mislabelling signal detected for the "tuna" category was mostly attributable to the samples sold as "bluefin tuna." To account for this fact, this category was then split into "bluefin tuna" and "tuna", although only 6 samples fall under the "bluefin tuna" name. This procedure ensures that most categories have a large enough sample size for statistical analysis while being representative of the French market.

Note that for reading convenience and international comparison, the French fish names have been translated into their English equivalent when available and used throughout this study (Table 1). Some (such as "colin," referring to a broad category of white meat species) could not be translated, and were kept in their original form. Furthermore, as the French vernacular names relate to the local naming traditions, they might not designate the

same species as in countries using the English equivalent. For instance "albacore" refers in French to *Thunnus albacares*, while in English it refers to *Thunnus alalunga*.

## Statistical analysis

The substitution status of the samples was analysed as a binary variable using a generalized linear model with a binomial error distribution and a logit link function. The type of protocol, retail type, species category and type of product sold were included as explanatory variables, with interactions.

After removal of the non-significant interactions and variables, Tukey Honest Significant Differences were calculated from the final model.

The influence of the price was investigated in a separate analysis for a subset of 156 samples for which the information on the price was available and could be expressed in €/kg. The substitutions were modelled as above with the price, the retail conditions and the type of shop as independent variables, with interactions.

All the confidence intervals ($\alpha = 0.05$) were calculated using Wilson's method. The statistical analysis was performed with R (*R Core Team, 2013*) and both the script used and the original data file are available on Figshare (http://dx.doi.org/10.6084/m9.figshare.978485).

## Supplier interviews

A follow-up investigation was performed for samples for which mislabelling was detected. Retailers were met in person or contacted over the phone. Interviews started by presenting the study, the sample purchased in the shop and explaining that we detected a mismatch between the molecular determination and the label. The supplier was then asked several questions (Fig. S1) to determine whether the substitution was intentional; and if so, what motivated the substitution.

# RESULTS

## Sampling and sequencing

We collected 291 samples using the FL protocol, out of which 276 could be sequenced. We obtained 172 long sequences, 97 intermediate length and seven short sequences. Fifteen samples (5,16%) could not be amplified at all, a failure rate comparable to other studies of this type (*Hanner et al., 2011*; *Cawthorn, Steinman & Witthuhn, 2012*). These included nine ready-made dishes, the single canned sample present in the sampling, two smoked fillets, and three restaurant dishes, all sources that are expected to show some DNA degradation.

All 114 TE samples provided sequences (45 long sequences and 69 short).

Both datasets together added up to 390 sequences. Nineteen of these were then removed because important information was missing, or doubts remained on the quality of the collected data. The final dataset therefore included 371 samples.

## Molecular identification

Identifications using the sequence similarity, the position in the BOLD distance trees and the verification of species-specific sites in the sequences gave congruent results. For 90% of the samples, the similarity with sequences present in BOLD was high: between 99.19%

and 100%. Almost all species included in the study were represented in the BOLD by barcode clusters that are single, cohesive, and non-overlapping with other species clusters (*Hanner et al., 2011*), a prerequisite for good identification. Most groups also had relatively high interspecific divergences in BOLD even with the most closely related species, making assignation to a single species straightforward. Tree-based identifications placed most samples within large clusters with the same identification (grade A identification according to *Costa et al., 2012*). All these can therefore be considered as non-ambiguous, high reliability identifications. It was, however, less straightforward for three particular groups (European sea bass, tuna and rockfish), although the end result can be considered reliable.

The first, "bar" (European sea bass, *Dicentrarchus labrax*), is represented by three divergent clusters in BOLD. Part of our samples are almost identical to samples from the UK and Spain; the rest are almost identical to samples from Turkey and Portugal. These two groups of sequences diverge by 2.5%–3% from each other. Therefore, *D. labrax* samples had a grade C identification (*Costa et al., 2012*) if considering only the sequences in BOLD. However, one of these groups of sequences is identical (or with one base divergence) with the highly reliable FDA208 *Dicentrarchus labrax* sequence in the Reference Standard Sequence Library for Seafood Identification of the FDA, and the reliability of the identification for the second group is also supported by reference sequences from independent datasets in BOLD.

Conversely, tuna species presented little interspecific divergence and are difficult to identify using similarity or clustering-based methods (*Lowenstein, Amato & Kolokotronis, 2009*; *Vinas & Tudela, 2009*). However, some species had clear shared sites in the sequences (*Lowenstein, Amato & Kolokotronis, 2009*) that could be used to group the sequences. This was the case for *Thunnus thynnus* ("thon rouge"), *Thunnus alalunga*, and our samples of *Thunnus obesus*. *Thunnus albacares* sequences in BOLD formed a cluster with more variability. This cluster also contains sequences identified as other *Thunnus* species. As these other species usually group in distinct BOLD clusters, the most probable explanation of their placement in the *T. albacares* cluster is misidentification of some of the sequences in BOLD. Therefore, sequences falling into this cluster were attributed to *T. albacares*.

Third, the "sebaste" (rockfish) sample was embedded within a cluster of related rockfish sequences from BOLD. This cluster also contained sequences from other closely related species. Therefore, only genus-level identification was possible. The same problem had already been encountered by *Wong & Hanner (2008)* and *Hanner et al. (2011)* on the same genus.

Two pairs of mixed samples could be identified (FL0084 and FL0085, FL1263 and FL1266). Each sample of the pair originate from the same collection event (same day, same place, same collector) and the molecular identifications are exactly switched. As the most probable explanation is accidental exchange by the collector, these samples were switched back and kept for analysis.

The summary of the commercial names and species determination is presented in Table 1. A total of 42 species were identified. This number far exceeds the number of targeted species because (i) several species can be sold under a given commercial name,

**Table 2 Substitution cases.** The "thon rouge" (bluefin tuna) category account for 5 out of the 14 substitions observed in our sampling ($n = 371$), although it contains only 6 samples.

| | Sample | Commercial name (latin name if indicated) | Dataset length | Similarity with BOLD sequences | Molecular identification | Origin | Zipcode |
|---|---|---|---|---|---|---|---|
| **Tuna** | | | | | | | |
| | TE14 | Thon rouge | L | 100.00% | *Thunnus albacares* | Fishmonger filet | 75 |
| | TE32 | Thon rouge (*Thunnus thynnus*) | S | 100.00% | *Thunnus obesus* | Fishmonger filet | 75 |
| | TE109 | Thon rouge | L | 100.00% | *Thunnus albacares* | Fishmonger filet | 75 |
| | FL0183 | Thon rouge | L | 100.00% | *Thunnus albacares* | Restaurant | 50 |
| | FLID1031 | Thon rouge | L | 100.00% | *Thunnus albacares* | Fishmonger filet | 75 |
| | FLID116 | Thon albacore | M | 100.00% | *Thunnus obesus* | Supermarket fresh filet | 75 |
| **Cod** | | | | | | | |
| | TE63 | Cabillaud | S | 100.00% | *Melanogrammus aeglefinus* | Fishmonger filet | 75 |
| | TE112 | Cabillaud (*Gadus morhua*) | L | 99.69% | *Melanogrammus aeglefinus* | Fishmonger filet | 75 |
| | TE190 | Cabillaud | L | 99.85% | *Melanogrammus aeglefinus* | Fishmonger filet | 75 |
| | FL0196 | Cabillaud | M | 100.00% | *Pollachius virens* | Restaurant | 75 |
| | FL0572 | Cabillaud | M | 100.00% | *Melanogrammus aeglefinus* | Supermarket fresh filet | 77 |
| | FL0963 | Cabillaud | L | 100.00% | *Melanogrammus aeglefinus* | Supermarket fresh filet | 76 |
| **Sole** | | | | | | | |
| | FLID089 | Sole | L | 99.19% | *Cynoglossus senegalensis* | Supermarket fresh filet | 77 |
| **Red Porgy/seabream** | | | | | | | |
| | FL0007 | Pagre | M | 100.00% | *Gadus morhua* | Restaurant | 75 |

(ii) the substitutions increase the number of species detected, and (iii) the collectors sampled more species than targeted.

## Species substitution

Among the 371 samples analysed, we found 14 cases of species substitution, representing a rate of 3.7% (CI [2.2–6.4], Table 2). We found substitutions for the following five fish species: bluefin and yellofin tuna, cod, sole and red porgy/seabream (see Table 2). As expected, most of these products were substituted for species with a lower market value. Five substitution cases were observed for bluefin tuna, although this species is represented by only 6 samples. The substitution rate for this species is 83%, with a confidence interval of 36%–99%.

The species representation was largely uneven, with the top five species totalling 67% of the samples and none of the remaining categories containing more than 18 samples (Fig. S2). The samples were more evenly distributed among retail types, with 74 samples from fishmongers, 100 from restaurants and 197 from supermarkets.

No effect of the protocol (FL or TE) on the rate of substitution was detected in the full model. They were thus pooled. The "species" variable has an impact on the rate of misla-belling ($p < 0.001$, Fig. 1). Post-hoc testing indicates that this is due to the "bluefin tuna"

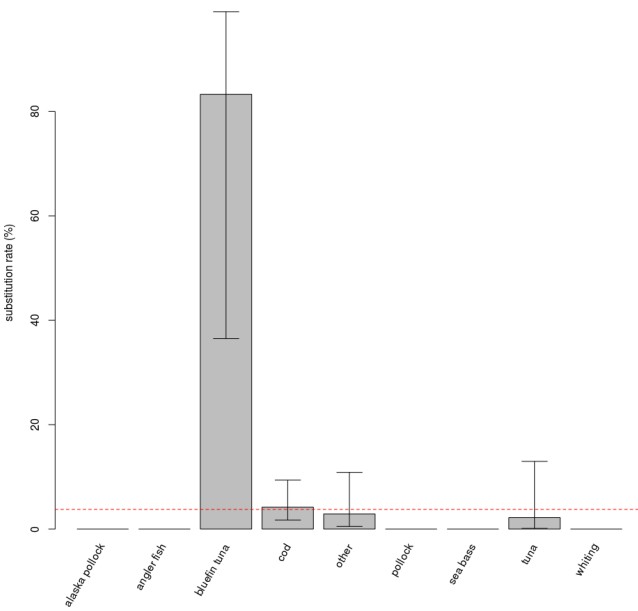

**Figure 1 Substitution rates for different commercial name categories.** Species categories with more than 10 samples collected have comparable, low substitution rates; substitutions were observed in only three of the categories. Bluefin tuna displays an exceptionally high substitution rate, and was separated from other tuna species in the figure and in analyses, despite a very low number of samples ($n = 6$). Error bars show the 95% confidence interval. The red dashed line is the average substitution rate observed for the entire dataset.

category being significantly different from the three categories with the largest number of samples, i.e., "cod", "other" and "tuna" (respectively $p = 0.004$, $p = 0.006$, $p = 0.012$).

The different modes of retail also show a marginally significant difference among them ($p = 0.085$), as species substitution was found only for products sold as fresh fillets or as restaurant meals (Fig. S3).

No effect of the price on the probability of species substitution was observed (data not shown).

## Participative collection

The whole TE sampling and part of the FL sampling were done by volunteers. The sampling effort is very broadly distributed, with the top 3 collectors contributing to 36% of the sampling efforts (with respectively 72, 49 and 15 samples), while 75% of the collectors contributed one or two samples each.

## Supplier interviews

Out of the 14 cases of mislabelling identified, four fish retailers, two restaurant owners and four supermarket executives were contacted in person or over the phone. In five cases, they responded positively up to the third question, acknowledging an intentional substitution. Two non-exclusive reasons were given: (i) increased gains due to the price difference between the two species and (ii) replacement of a highly demanded species by an easily available, less considered species.

## DISCUSSION

This study is the first assessment of fish mislabelling in France, the largest seafood market among European countries. The samples presented observed an overall rate of substitution of 3.7% CI [2.2–6.4], which is low compared to the rates reported for most other countries (Table 3). This rate might partly be a consequence of our broad sampling scheme, which included multiple sample sources and supply lines. Contrary to fresh fillets or restaurant meals (both known to be prone to species substitution), industrial products like deep-frozen fillets or ready-made meals have been shown to present either very low substitution rates, as in the UK (below 1.5% for fish fingers, *Huxley-Jones et al., 2012*) or much higher ones (above 30%, *Di Pinto et al., 2013*; *Garcia-Vazquez et al., 2011*) depending on the country. We found no case of substitution among these products in our sampling, suggesting that the situation in France is closer to that of the UK. The low rate observed for these products might thus have decreased the overall substitution rate in our study.

The difference in substitution rates between countries might be the result of many social and economic factors, such as the rate of control by government agencies or the length of the supply chain, but few of them have been specifically investigated. One notable exception is the case of Ireland, where media attention led to an improvement in substitution rates (*Mariani et al., 2014*). However, due to the lack of older data for France, it is not possible to know whether there was a similar effect and if the rates have changed over time.

Fish species substitution rates have also proved to be highly variable among European countries (Table 3). However, the sample acquisition method is not standardized across studies, and comparisons between the observed rates must be undertaken with care. Notably, species availability, prices and consumers preferences differ between geographic areas and limit comparisons across studies and countries. Cod is probably the species that can best be compared, as it is represented in most studies by the largest number of samples. Our cod sampling is similar in size to the sampling of several cod-centered publications (*Di Pinto et al., 2013*; *Miller & Mariani, 2010*; *Miller, Jessel & Mariani, 2012*). In comparison, our substitution rate is one order of magnitude lower than in Italy (*Di Pinto et al., 2013*) or Ireland (*Miller & Mariani, 2010*; *Miller, Jessel & Mariani, 2012*, Table 3), and similar to that of the UK (7.4% in *Miller, Jessel & Mariani, 2012*). This low rate for France is very encouraging, but the origins of the differences between countries remain to be investigated. They might provide clues for a better resource and market management.

While the substitution rate is low, there is a consistent pattern: a species is replaced by one with a lower commercial value. This pattern is also observed in other countries and hints at economic motives. We did not observe substitutions of species claimed to be sustainable by species that are not, whereas in the UK Pacific cod replaces Atlantic cod (*Miller, Jessel & Mariani, 2012*). Our dataset contains very few samples of the Pacific species *Gadus macrocephalus*, although it is legally acceptable under the widely-used commercial name "cabillaud."

Since the number of samples per species shows high variability (Fig. S1) and the substitution rate is low, we observed few cases of mislabelling per fish name category, preventing detailed comparisons between them. However, we detected an effect of the

**Table 3 Substitution rates observed in similar studies.** These studies all used molecular identification to estimate the rate of species substitution.

| Investigated country | Substitution rate | Nb of sequences | Taxonomic focus | Origin[a] | Type[b] | Marker | Reference |
|---|---|---|---|---|---|---|---|
| **EU** | | | | | | | |
| Ireland | 19.00% | 111 | diverse | F, S, R | Fl, Fr, P, Rd | COI | *FSAI, 2011* |
| Ireland | 25.00% | 156 | cod | F, S, F&C | Fl, Fr, Rd | COI | *Miller & Mariani, 2010* |
| Ireland | 28.20% | 131 | cod | F, S, F&C | Fl, Fr, P, Rd | COI | *Miller, Jessel & Mariani, 2012* |
| Ireland/UK | na | 98 | Rajidae | F, S, F&C | Fl, Rd | COI | *Griffiths et al., 2013* |
| UK | 7.40% | 95 | cod | F, S, F&C | Fl, Fr, P, Rd | COI | *Miller, Jessel & Mariani, 2012* |
| UK | <1.5% | 142 | diverse | S | P | COI | *Huxley-Jones et al., 2012* |
| Italy | 32.00% | 69 | diverse | F, S | Fl, Fr | COI & Cytochrome b | *Filonzi et al., 2010* |
| Italy | 77.80% | 59 | Mustelus sp. | F, S | Fl | COI | *Barbuto et al., 2010* |
| Italy | 56.36% | 110 | cod | S | Fl, P | COI | *Di Pinto et al., 2013* |
| Italy | 20.00% | 18 | diverse | Port authority | P | COI & Cytochrome b | *Cutarelli et al., 2014* |
| Spain | >20% | 40 | Hake | S | Fr, P | Mt Control region SNPs | *Machado-Schiaffino, Martinez & Garcia-Vazquez, 2008* |
| Spain & Greece | >30% | 279 (93*3) | Hake | S | Fr | 5S rDNA, CytB RFLP | *Garcia-Vazquez et al., 2011* |
| France | 3.7% CI [2.2–6.4] | 371 | diverse | F, S, R | Fl, Fr, P, Rd | COI | Present study |
| **Non-EU** | | | | | | | |
| Japan | 8.00% | 26 | Tuna | F, R | Fl, Rd | COI, Mt Control region, ITS 1 | *Vinas & Tudela, 2009* |
| South Africa | 21.00% | 248 | diverse | S, F | Fl, Fr, P | COI | *Cawthorn, Steinman & Witthuhn, 2012* |
| South Africa | 50.00% | 174 | diverse | R, F | Fl, Fr | 16S rDNA | *Von der Heyden et al., 2010* |
| Canada | 41.20% | 236 | diverse | F, S, R | Fl, Fr, Rd | COI | (*Hanner et al., 2011*) |
| US | 32,35% | 68 | Tuna | R | Rd | COI | *Lowenstein, Amato & Kolokotronis, 2009* |
| US | 11.00% | 99 | Salmon | R, S | Fl | COI | (*Cline, 2012*) |
| US & Canada | 25.00% | 90 | diverse | F, R | Fl, Rd | COI | *Wong & Hanner, 2008* |

**Notes.**

[a] F, Fishmongers; S, Supermarkets; F&C, Fish and chips; R, Restaurants.

[b] Fr, Frozen; P, Prepared dish (includes fishfingers and battered); Fl, Fillet; Rd, Restaurant dish.

"commercial name" variable on the substitution rate. This effect was mostly due to bluefin tuna. Market issues are particularly important for bluefin tuna because of its conservation status. We found this species to be highly mislabelled, with 5 out of 6 samples being substituted (i.e., 83% CI [36–99]), which stands in sharp contrast with the low substitution rate over the whole sampling. Contrary to some other studies (*Wong & Hanner, 2008* for instance), however, bluefin tuna was substituted with other tuna species and never with unrelated species. Moreover, for 16 samples collected in sushi restaurants the waiters replied upon enquiry that the tuna sold was bluefin tuna, which was never the case (data not shown but included in the BOLD repository). Although we excluded these samples

**Peer**J

from our analysis because the menu was not precise enough, this shows an absence of care or knowledge in the usage of this commercial name.

The catches of this species were largely debated, and the presentation of this issue in the media was positive in influencing fisheries management (*Fromentin et al., 2014*). They probably make up the most lucrative fisheries in the world, driven by strong demand from the Japanese market (80% of the global catches) (*European Commission, 2009*). This commercial importance led to severe overfishing during the 1990–2000s, with estimates of stock declines of 72% in the Eastern Atlantic, and of 82% in the Western Atlantic (*ICCAT, 2009*). International concerns over the species survival culminated in 2009 with the proposal to protect bluefin tuna under the UN Convention on International Trade in Endangered Species (*CITES, 2008*), which was eventually rejected. Since then, the implementation of strengthened management measures resulted in reductions in catches and fishing mortality rates, indicating that the species may be slowly recovering (*Fromentin et al., 2014*; *ICCAT, in press*).

There are at least two plausible explanations for the high mislabelling rate of this species. First, bluefin tuna is called "red tuna" ("thon rouge") in French. This might confuse waiters and customers, as fresh tuna meat is reddish. Therefore, any raw tuna meat can appear as "red." Second, as highlighted by its conservation issues, this species is considered on the French and other markets to have a high quality meat and might appear more attractive to the customers. These two factors might have acted together: the high demand of customers pushing the retailers to take advantage of the confusing French name of the species.

The probability of substitution might also be influenced by the retail type, although this trend is not statistically significant in our study. This might be due to the small number of substitutions observed ($n = 14$), but several lines of evidence suggest that there might be a real difference. First, we found no case of substitution in industrially processed food like prepared meals ($n = 67$) or deep-frozen fillets ($n = 33$, Fig. S3). For a species heavily used by the industry like the Alaska pollock, we observed no case of substitution despite a significant sample size ($n = 33$). Second, 10 out the 14 substitutions were investigated by interviews with suppliers. In five cases out of ten, the people responsible for the last step before the fish reaches the consumer admitted intentional substitution for increased profit or consumer expectation reasons, in agreement with studies in other countries (as reviewed in *Jacquet & Pauly, 2008*). There are no such last steps for the prepared meals and deep-frozen fillets.

Our study was made possible by the involvement of dozens of volunteers throughout France. Citizen science has emerged in the last decades as a way for scientists to have access to large datasets extending the studies in space and time (*Hochachka et al., 2012*) or to have humans performing tasks that computers cannot, as exemplified by the Galaxy Zoo (*Clery, 2011*) or FoldIt (*Cooper et al., 2010*) projects. Some authors have distinguished different types of collaborative work between scientists and citizens, depending on the involvement of citizens in the research tasks (*Cooper et al., 2007*). Our study had a mixed type of research management. It was initiated by non-scientists involved in controlling the economic use of natural marine resources. They were then joined by scientists to

ensure that the study will meet the stringent criteria of peer-reviewed science, a model referred to as "participatory action research" by *Cooper et al. (2007)*. Finally, volunteers were recruited to enlarge the dataset, following a research model more common in citizen science. Our study has mobilized two types of citizens: the initiators of the study, who actively participated in all the research tasks; and the collectors, who enabled the large scale of the study by collecting samples.

Involvement of non-specialists can represent a problem for the reliability of the sampling. For instance, if there were mistakes by inexperienced collectors, the most probable effect is additional "substitutions" recorded (false positives), with the result being an over-evaluation of the real number. It was not possible to check the whole sampling process for each collector, but we checked the samples of each collector to know whether they sampled more substituted products than average. We also carefully checked each substitution case. All of them came from different retailers, except for one supermarket line for which two substitutions were detected from two different species by two different samplers in two different areas of the country.

However, the very low substitution rates found in this study also confined the potential problem. If over-enthusiastic collectors focused on places where they expect to find substitutions, or made mistakes in the collecting, they might have increased the number of substitutions compared to a non-biased sampling. While we cannot exclude such a bias, it would mean that the rate of substitutions is actually even lower than described here. However, at least 5 of the 14 cases were corroborated by the persons responsible for the substitution themselves. They form therefore a reliable minimum, with a maximum at 14 (still very low compared to most other studies already published, see Table 3), as all our possible biases would tend to increase the number of substitutions recorded.

The congruent results for the two sets (TE and FL) collected and sequenced independently also speaks in favour of the reliability of the sampling.

## CONCLUSION

This study was designed to cover a large part of the French fish products market, as we aimed at estimating mislabelling rate over multiple product types. Compared to substitution studies in other countries, we observed a low substitution rate. Detailed analysis reveals two trends that need further investigation.

First, some species appear to be more often substituted than others, with multiple cases observed for cod and bluefin tuna, as was also established in other countries and studies. The substitution rate on bluefin tuna was especially high, which might be linked to the public debate on this species. Specific studies focused on this species would be needed to confirm this finding in France and in other countries.

Second, the rate of mislabelling seems to differ between supply chains. We detected no mislabelling in industrial products, while several restaurant owners or fishmongers acknowledged intentional substitution. This suggests that substitution is more important at the end of the supply chain and that control efforts must be directed at this level.

Despite limitations in a few taxa, DNA barcoding based on the COI sequence provided fast, efficient and unambiguous identifications for most of our commercial fish samples even when only a short fragment was used, in line with previous studies (*Meusnier et al., 2008*). The BOLD hosted dataset gave a resolution superior to the one in GenBank, and the tools available with the database permit an easier evaluation of dataset quality and homogeneity.

Although the substitutions appear infrequent compared to other studies and concentrated on some species and retail types, improvements can be made to increase the reliability of the market. In particular, the scientific names were indicated for only a low proportion of the samples at sale. In France, like in other countries, legislation on labelling differs between restaurants, fresh sales and deep-frozen fish. For some groups like rays or tuna, the authorized commercial names cover a large number of species, including species with serious conservation concerns. In such cases, there is no way for knowledgeable consumers to choose according to sustainability criteria, and controls could be improved without systematically resorting to testing. We join *Miller, Jessel & Mariani (2012)* and *Jacquet & Pauly (2008)* in their call for more precise and informative labelling and hope that publicly available data will help citizens, through media attention, to push for this type of change as exemplified by recent progress in Ireland (*Mariani et al., 2014*).

## ACKNOWLEDGEMENTS

We thank all our collectors for their great and thorough work, and Baptiste Carton and Alexia Velasquez for help with the lab work and the sequence management. Laboratory access and assistance was provided by the "Service de Systematique Moleculaire" of the Muséum National d'Histoire Naturelle (CNRS UMS2700 OMSI). We thank Dusan Misevic for commenting on earlier versions of this paper, and S. Von der Heyden, an anonymous reviewer, and the editor R. Toonen for their constructive criticisms. We also thank G. Lancelot and N. Schnell for their English and content corrections.

### Funding

Julien Bénard-Capelle was funded by the "COOPINFO" grant (number 10 BLAN 1724 01) from the French National Agency of Research to INSERM U1001. The study was funded by INSERM U1001 (sequencing), MNHN Département Systématique et Evolution (DNA extraction and amplification), BLOOM (sample collection), Oceana and Terra Eco (sample collection, DNA extraction and amplification, sequencing). The funders had no role in study design, data collection and analysis, decision to publish, or preparation of the manuscript.

### Grant Disclosures

The following grant information was disclosed by the authors:
French National Agency of Research to INSERM U1001: 10 BLAN 1724 01.
MNHN Département Systématique et Evolution.

BLOOM.

Oceana and Terra Eco.

## Competing Interests

Victoire Guillonneau and Claire Nouvian work for BLOOM, Nicolas Fournier works for Oceana Europe, and Karine Le Loët works for the journal Terra eco.

## Author Contributions

- Julien Bénard-Capelle and Agnès Dettai conceived and designed the experiments, performed the experiments, analyzed the data, contributed reagents/materials/analysis tools, wrote the paper, prepared figures and/or tables, reviewed drafts of the paper.
- Victoire Guillonneau, Claire Nouvian, Nicolas Fournier and Karine Le Loët conceived and designed the experiments, performed the experiments, contributed reagents/materials/analysis tools, reviewed drafts of the paper.

## DNA Deposition

The following information was supplied regarding the deposition of DNA sequences:
BOLD Project FSCF: FCSF001-14 to FCSF404-14.

## Data Deposition

The following information was supplied regarding the deposition of related data:
FigShare: http://dx.doi.org/10.6084/m9.figshare.978485.

## Supplemental Information

Supplemental information for this article can be found online at http://dx.doi.org/10.7717/peerj.714#supplemental-information.

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
