# Peer review of "Fish mislabelling in France: substitution rates and retail types"

_PeerJ, doi:10.7717/peerj.714_

## Round 0.1 · original submission · Major Revisions

· Academic Editor

Major Revisions

Overall both referees seem to agree that the manuscript is of interest to the readers of PeerJ and should be published upon satisfactory revision. However both also have a number of criticisms of the writing and make numerous suggestions for revision to clarify the text. Both referees were confused by the study design, which clearly needs to be corrected before the paper can be accepted for publication. Additionally, both referees state that they have issues with the experimental design because it is easily biased (even unintentionally) and are therefore unconvinced that the statistical analyses and comparisons to previously published works are valid. If you want to include these comparisons, I feel that you need to include an explicit acknowledgement of this issue in the manuscript. Finally, the second referee has extensive suggestions for improving the manuscript, and although many are straightforward issues with spelling or grammar, they are important for the final revision of the manuscript.

Given the issues of both clarifying the experimental design and acknowledging the limitations and potential biases of the sampling design are serious and need to be corrected before the manuscript can be accepted for publication, I have labeled this as a major revision. I look forward to seeing your revised manuscript.

·

Basic reporting

This manuscript meets the PeerJ standards. Although the manuscript is straightforward and does not add immensely to the field of 'seafood forensics', it does provide a decent overview of European, but especially policy context and consumer background. I do think that three of the four figures are superfluous, namely 1, 3 and 4. Good descriptions of this is given in the text. In their place, I would rather see a tree or the like, that demonstrates the findings, rather than sampling protocol.

Another point i would like to make pertains to conflicts of interest when including not only citizen scientists in such studies, but also journalists. A large number of samples were collected by members belonging to these groups and I am always wary about their independence. What if they purposefully targeted establishments, which might, even if only by hearsay, have already been implicated in some sort of fraud? It only takes a casual "I went there last night and I don't think the cod they sold is really cod", to bias the sampling. However, the study has now been carried out, but it is something that the authors need to be aware of for any future studies along these lines.

Experimental design

This was generally sound. In L153 the authors say that they constructed distance trees, but they do not mention which program they used for this. This needs to be added. Also, how did they evaluate the relationships between known samples and those from their study? It should be done with bootstrap supports.

I was also a bit confused with the total sample numbers? In the abstract they mention that they sequenced 390 samples, but there were some that did not amplify. It was unclear whether these were included in the 390 samples, or not. Please make sure the total sample numbers are correct throughout. In L277 you say 371 were kept for analysis.

Validity of the findings

They are sound - no comments.

Additional comments

Please go through the MS carefully and check for grammatical errors and typos. I came across several and I have made suggestions on how to fix certain parts. However, I stingily recommend that a native english speaker goes through the MS and sorts out some of the harder to understand parts.

L20: full stop missing
L33: "Fish are among the last wild harvested species" - well I do not agree with that. There are MANY species that are still wild harvested, perhaps not in Europe, but certainly in many other parts of the world. I would delete this sentence as it is not truthful.
L36: please watch out for floating commas - there are a few around
L37: actually, numerous fisheries were already collapsing in the late 60's and during the 70's.
L39: hw about 'geographically far removed' rather than distant. This sentence reads awkwardly
L39: are aquatic foods really the most widely traded types? More so than wheat soya and rice?
L63: here you should give some context as to why France was chosen as a study place. Further on in the MS you give context as to how many tons are consumed by year, as well as a market value and I suggest that you move this here to give some background and rationale for your study.
L64: required sounds better than needed
L114: English with capital E
L238: part of samples are, not is
L286: do you mean comparable rather than compatible?
L315, 329: floating :
L362 onwards - I really liked this section
L370 and elsewhere: the convention is to spell out number names less than ten and >ten as numerals. Check the journal requirements on this
L376: meanS
L426: this sentence does not read well. Try something along the lines of "identifying individuals to the population level, rather than just species".

Table 3 and elsewhere: in one of our studies (von der Heyden et al 2010), we found substitution rates of up to 80% for some species, and that more than 50% of the fillets we tested were substituted. This is higher than some of the rates in this table. I also think that a useful addition here would be to include marker type so that the reader can get a good overview of the studies.

Reviewer 2 ·

Basic reporting

After a thorough reading of the manuscript, I feel as though the writing is currently not of a standard that is high enough to merit publication within PeerJ. Main concerns include errors in spelling and grammar throughout. Sentence and paragraph structure is also a concern, particularly within the Introduction. It is difficult to follow the flow of the writing at times as themed groups of ideas and points are not clearly clustered within paragraphs (e.g., again, the introduction needs a lot of work in this respect). It feels as though generally, the writing within the whole manuscript is not organized well and lacks a clear flow through ideas. It would be useful for the writers to examine the paper again, produce an outline of the core ideas that need to be communicated and structure the writing around this outline. The paper also reads in a strange way in that the quality and style of the writing does not seem to be consistent. I would suspect that this could be the result of multiple authors each writing a separate section. To improve this, a single (most experienced) writer should go through the entire paper and revise so that the style, quality and flow of ideas are consistent.

Specific notes:

Abstract:
-needs a lot of work with structure, clear components of introduction, methods, results and implications of work.
-the abstract is also inconsistent in the level of detail provided (e.g., little detail on methods (e.g., what species were sampled?), specific results given (e.g., for blue fin tuna), but general results not provided (e.g., how many fish samples were mislabeled??).
-why were the results 'remarkable?'?
-very little mention of 'investigation' methods and results throughout the paper. I would omit this from the paper as it appears as though the results of this work have already been published elsewhere (Le Loet, 2014)?
-is the main focus of the paper the effectiveness of citizen science or the mislabeling of seafood in France? Pick one and focus on this both in abstract and throughout paper. If the focus of the paper is testing the effectiveness of citizen science, how has this been evaluated?

Introduction:
-general spelling, grammar and sentence structure errors or issues throughout.
-throughout the paper, 'Jacquet' is incorrectly spelled 'Jaquet' in citations.
-make sure abbreviations spelled out in words the first time they are used in text.
-no need to list all publications on seafood mislabeling, refer to a few examples instead.

Materials and methods:
-not enough basic details provided (e.g., not clear how many samples were collected, how they differed from each other (how many commercial names, how were they collected, how were they prepared)? If all of these details were provided, it wasn't presented in an organized fashion and so the most important details were not effectively communicated. This needs to be fixed.
-what is 'semi-participative'? More explanation needed here
-'transformed' fish? Need more description here as to what is meant by this.
-not necessary to provide details of primers in text - if felt needed, it should be presented in a more attractive fashion, not just stuck within the text.
-better subheadings may help to bring clarity to this section - perhaps to separate explanations of how the two batches of samples were collected and treated differently
-some awkward phrasing and sentence structure throughout - overly confusing when it doesn't need to be
-again, far too much detail on some components of methods (e.g., molecular and statistical techniques), but far too little on other components (e.g., follow up investigation)
-if work from follow-up investigation was already published (Le Loet, 2014), remove from methods and results and only refer to this work in the discussion.
-if work from follow-up investigation has not been already published, much much more detail is needed on study design (semi-structured interviews? how was data analyzed?? etc., etc., etc.)

Results:
-some of this should be in methods also (e.g., number of samples sequenced). -again, not clear how many samples were used in this study - abstract said 390, not clearly given in methods?, and 291, or 371 in results? This is very confusing.
-again, some parts of results overly detailed and basic, most important details are either ineffectively communicated or absent. Either way, the reader is left a bit confused.

Discussion:
-spelling, grammar and sentence structure issues throughout
-FAOSTAT spelled 'FOASTAT'??
-again, confusing as to what the main things that are meant to be communicated are - citizen science or mislabeling in France? Separate subheadings would be helpful to separate thoughts for two issues discussed but more emphasis needs to be given to one or the other.
-needs a separate conclusion section.
-needs more thoughts on policy recommendations, insights, important implications from work. What does this study tell us that we didn't already know???

Experimental design

The outline of experimental design was confusing. For this reason, it is difficult to make a judgement on whether the work is scientifically sound. The writing in both the methods and results section needs to be better organized, streamlined, and condensed. Remove superfluous details and make sure the important details (e.g., sample size, sampling design) are clearly described. I cannot tell what the sampling design was - what is 'semi-participative' collection? Were samples randomly collected by geographical area or commercial name? What were the instructions that were provided to collectors? How were collectors recruited? From the information given, it cannot be determined whether sampling was biased and I suspect it was... Collectors likely gathered samples they 'suspected' were mislabeled. The authors even allude to this in the discussion - "Collectors were generally enthusiastic out of curiosity for the possible substitution cases they might have been the victim of". If this is true, the data is not independent and it is probably not a good idea to apply statistics or make comparisons to other studies in the way the authors have done. The authors make comparisons to other studies in the discussion, but also mention that the design of their study differed to others because it was "designed to sample a large array of products". Again, it is not clear how the study was "designed" to specifically cover this sampling pool, and comparisons to other studies are not very informative because, as the authors pointed out, other studies have looked more specifically at particular species or types of seafood. Again, the research question is not clearly defined. Are the authors testing the effectiveness of citizen science, or are they evaluating the rate of fish species substitution in France? Either way, as presented, it is not clear whether the design of the study is sufficient to draw scientific conclusions on either of these issues.. although if the methods and results sections were improved, this view might change.

Validity of the findings

Comments for the above section also apply here.

Additional comments

I feel that this could be appropriate for publication but substantial improvements are needed. First, the writing needs to be improved so that most importantly the research questions, methods and results can all be clearly understood. After this has been done, the paper can then be evaluated for scientific validity.

---

## Round 0.2 · Minor Revisions

· Academic Editor

Minor Revisions

Both referees agree that the majority of their concerns were adressed in the revision and that this version is greatly improved from the original submission. The main contention remains the extrapolation of the results from your study to the overall marketing and generalizations beyond the scope of the sampling. The second referee, in particular, has a number of comments on the validity of the findings where the conclusions should be softened or applied with caveats. I feel these are appropriate and would like the authors to take them seriously in the revision. Both referees agree about the importance of the work and its value to the literature, so the work will stand without overstating the applicability or generality of the study. I expect that the authors should be able to report their work without overstating the conclusions and still have a solid paper of value to the field. Thus, it should not be difficult to address these comments and make the suggested revisions, and I look forward to seeing the revised manuscript.

·

Basic reporting

There are a few minor language issues remaining. I have listed some of these below, but strongly encourage the authors to carefully check before final submission as to make this the most easy to read and lucid manuscript possible.

L19 (and throughout): some of the sentence are short and rather clipped. See where you can add some of the shorter sentences together to improve flow. In L19 for example, I would start with 'However', in order to soften this sentence.
L19: in, not on many markets
L33: not just reliable but correct also.
L48: the middle sentence (Yet no results) is rather stuck in the middle here, so join with the previous using a yet.
L47: is France ranked 7th in the world? if yes, then clarify this here.
L51: greatly, rather than a lot.
L74: this reads rather awkwardly, suggest deleting 'for samples of a number'. Here I would also provide detail as to which are the most common commercial marine fish species. You hint at this later on, but I would suggest making it more explicit and easier to follow for the reader.
L82: the sentence ' it is the first of its kind for France' is stuck here. You need to move this up to the beginning of the paragraph and incorporate it there, rather than tagging it on at the end.
L105: April should be capitalised.

The discussion reads much better than the introduction. Try and follow what you did there for the introduction also.

Experimental design

This has been addressed and is fine.

Validity of the findings

Valid.

Additional comments

Thank you for addressing the comments in such detail. I think this makes a valuable contribution to understanding fraudulent seafood trades in Europe. Good luck if continuing with studies such as this.

Reviewer 2 ·

Basic reporting

In consideration of basic reporting, I recommend this paper should be altered according to the following instructions and revised considering the specific issues raised prior to publication (to be checked by editor):

Specific changes:

Abstract
- "assembled to date in a European country." not "in an European country."

Introduction
-review choices for citations (e.g., Line 13 - "aquaculture represents 40% of fish consumed in the world" - is this information from FAO's SOFIA?; Line 17 - Jacquet & Pauly 2008 is not a good reference supporting the statement relating to stock collapses)
-line 15 - does the management of wild marine resources really depend on traceability and labelling or is it just adversely affected by poor traceability and labelling?
-line 19 - two Cochrane et al. 2009 references in reference list - need to distinguish with 'a' and 'b'
-line 19 - available 'in' many markets not 'on'
-line 20 - along the 'supply chain' not 'line'
-line 28 - it doesn't seem like these issues should be described as 'risks'; maybe insert 'unknowingly' in front of 'purchase'.
-line 34 - observed a rise since when?
-line 44 - is the example you give inter-country variability or inter-product variability - how can you decide that is is one or the other, or both? - maybe add an 'and/or' somewhere here
-line 44 - this sentence is confusing and doesn't make sense to me, remove it or significantly rephrase - take out the word 'globally' to start (as this would suggest a much broad international study).
-line 45 - why is France the most important market? Largest quantity of seafood consumed here??
-line 47 - FAOSTAT data for 2009 is the most recent available? If not, use most recent reference
-line 48 - sentence shouldn't begin with 'Yet'
-line 48 - need a reference for statement that quality is regularly controlled
-line 55 - is there a more recent review available to reference here? Maybe (Griffiths, Andrew M., et al. "Current methods for seafood authenticity testing in Europe: Is there a need for harmonisation?." Food Control 45 (2014): 95-100)
-line 82 - "It is the first of its kind for France." - what is 'it'?
-line 105 - if commercial names were chosen in April, 2013, did sampling begin before sampling protocol was designed? (March 2013 - line 91)

Discussion
-line 332 - "contrary to in the UK" not "contrary to UK"
-lines 348 - 350 - needs citations
-line 374 - Jacquet & Pauly 2008 possibly a misleading citation to use -I don't believe field-based interviews were used in this work
-line 394 - use 'they' rather than 'he' or 'she' when referring to volunteers
-line 411 - don't use global when referring to this study (global suggests covering most countries of the world).

Conclusion
-line 416 - rephrase "...highlighting the public debate tension" as detecting mislabelling doesn't really highlight the public debate tension...?
-line 420 - might be a good idea to define 'industrial products' at first mention in paper (not sure where this was..)

Table 2
'Fillets' not 'filets'

General changes:
-check to make sure a consistent tense is used throughout the writing (past, or present, not both)
-check to make sure citation style is consistent (e.g., using 'and' as well as '&' in line 29 - Rasmussen and Morrissey 2008, Jacquet & Pauly 2008; comma instead of period in line 100 Aanesen et al, 2010; semicolon instead of comma in line 357)
-check to make sure consistent with ordering of citations either chronologically or alphabetically (e.g., list starting on line 37 doesn't seem to be in any particular order?)
-make sure consistent in use of words when using small numbers in text (e.g., line 228 "These included 9 ready-made... two smoked... 3 restaurant)
-check sentence structure throughout, a number of run-on and incomplete sentences throughout, e.g., line 398 starting sentence (and paragraph) with 'But.."

Experimental design

In consideration of experimental design, I recommend this paper should be altered according to the following instructions and revised considering the specific issues raised prior to publication (to be checked by editor):

-line 107 - salmon was 'initially excluded'? At what point was it included??
-line 117 - what is the justification for including only these products in the sampling protocol?
-line 142 - why were only 70 samples chosen to be sequenced in both directions?

Validity of the findings

In consideration of the validity of the findings, I recommend this paper should be altered according to the following instructions and revised considering the specific issues raised prior to publication (to be checked by editor):

-cannot claim study is representative of overall mislabelling rate in France as the sampling protocol specifically excluded products that were thought to have lower rates of mislabelling. For this reason, consider rephrasing a few statements in discussion, e.g., line 309 "overall rate of substitution", add 'in the samples collected'.
-line 311 - what does 'market-wide sampling scheme' mean? Your methods state certain products were purposefully excluded, so I would think this is not entirely a market-wide scheme... as with the comment before, consider this and review writing to make sure statements are accurate
-first paragraph of the discussion - the argument isn't well supported here. You speculate that low rate was because your study included a broader range of products in marketplace than other studies, then reference some of these same studies that included broad ranges of products to show they had lower rates of mislabelling.
-please read (Mariani, S., Ellis, J., O'Reilly, A., Bréchon, A. L., Sacchi, C. and Miller, D. D. (2014), Mass media influence and the regulation of illegal practices in the seafood market. Conservation Letters. doi: 10.1111/conl.12085) and refer to this in discussion, considering possibility that media has influenced lower rates following high global publicity in relation to seafood fraud (and other food fraud such as the horse meat scandal which came to light immediately before your sampling began in 2013).
-line 430 - would requiring scientific names on labels increase the reliability of the market - how?? Doesn't it just provide more information to consumers to make their decision, but the product may still be mislabelled? Include additional recommendations in your conclusion such as a suggestion to improve enforcement of labelling laws, implementation of monitoring schemes that integrate DNA testing, etc.

Additional comments

Commendable job in revising manuscript, but please make recommended changes and revise taking into consideration suggestions made and issues raised. If this is done sufficiently, as evaluated by the editor, I recommend publication.

---

## Round 0.3 · accepted · Accept

· Academic Editor

Accept

Thank you for your further revision of the text. I am now satisfied that you have addressed the issues raised by the referees, and am happy to recommend your manuscript move into publication.